In vitro histomorphometric comparison of dental pulp tissue in different teeth

Guerrero-Jiménez Marytere 1
http://orcid.org/0000-0001-8003-7716 Nic-Can Geovanny I. 1 2
Castro-Linares Nelly 1
http://orcid.org/0000-0002-9114-1807 Aguilar-Ayala Fernando Javier 1
Canul-Chan Michel 3
Rojas-Herrera Rafael A. 4
Peñaloza-Cuevas Ricardo 1
Rodas-Junco Beatriz A. 1 2 beatriz.rodas@correo.uady.mx barodasju@conacyt.mx
1 Laboratorio Traslacional de Células Troncales de la Cavidad Bucal, Facultad de Odontología, Universidad Autónoma de Yucatán , Mérida, Yucatán , México
2 CONACYT-Facultad de Ingeniería Química, Universidad Autónoma de Yucatán , Mérida, Yucatán , México
3 Facultad de Ciencias Químicas, Universidad Veracruzana , Orizaba, Veracruz , México
4 Facultad de Ingeniería Química, Universidad Autónoma de Yucatán , Mérida, Yucatán , México
Maddi Abhiram
Electronic publication date: 2019 Dec 6
Publication date: 2019
Volume: 7
Electronic Location ID: e8212
Received 2019 May 8; Accepted 2019 Nov 14
Copyright: © 2019 Guerrero-Jiménez et al.
Copyright year: 2019
Copyright holder: Guerrero-Jiménez et al.
License: This is an open access article distributed under the terms of the Creative Commons Attribution License, which permits unrestricted use, distribution, reproduction and adaptation in any medium and for any purpose provided that it is properly attributed. For attribution, the original author(s), title, publication source (PeerJ) and either DOI or URL of the article must be cited.
License URL: https://creativecommons.org/licenses/by/4.0/

Keywords: Dental pulp, Stem cell, Histomorphometric, Supernumerary teeth

Funding: Faculty of Chemical Engineering and Faculty of Dentistry, Autonomous University of Yucatan This work was supported by the Faculty of Chemical Engineering and Faculty of Dentistry, Autonomous University of Yucatan. The funders had no role in study design, data collection and analysis, decision to publish, or preparation of the manuscript.

==============================
Background

Dental pulp (DP) represents an accessible and valuable source promising of stem cells for clinical application. However, there are some disadvantages associated with the isolation of dental pulp stem cells (DPSCs), which include the size and weight of the pulp tissue needed to yield sufficient cells for culturing in vitro. Therefore, the objective of this study was to compare in vitro histomorphometry of DP from permanent (premolars, third molar), supernumerary and deciduous teeth of patients between 5 and 25 years old with regards to weight, length, width and the cell density in the four regions of the DP in order to obtain quantitative parameters in a tissue that represents a valuable source of stem cells.

Methods

DPs were obtained from 10 central incisors deciduous, 20 permanent teeth (10 premolars, 10 third molars) and 10 supernumeraries (six mesiodents and four inferior premolar shapes). The pulps were carefully removed, and the entire tissue was weighed. The pulp length and the width were measured with a digital Vernier caliper. The cellular density analysis was performed according to the four regions of the DP (coronal, cervical, medial and apical) in histological slides using photography and the ImageJ® program for quantification.

Results

The Pearson correlation test revealed that DP weight among different types of teeth is correlated with age in male patients. A significant positive correlation was noted between length and width of the DP with age in both genders. The mean DP weight for supernumerary and third molar teeth was greater than deciduous and premolar teeth. Finally, the histological analysis showed that the coronal and apical portions of DP in supernumerary and premolar teeth have the highest cell density.

Conclusions

The DP of supernumerary teeth has quantitatively the best morphometric parameters and cell density comparable with the quality of DP obtained from deciduous teeth.

Introduction

Dental pulp (DP) is an innervated, highly vascularized soft tissue that provides vitality to the tooth (Rodas-Junco et al., 2017). DP is located inside each primary or permanent tooth, and its main functions include the generation of dentin and maintenance of its biological and physiological vitality in response to traumatic injuries, physical stimulus or bacterial infections (Marrelli et al., 2018; Ravindra et al., 2015; Tatullo et al., 2015). The regenerative function of DP suggests that it contains odontogenic progenitor cells or stem cells that are involved in the regeneration process. In this context, DP has drawn attention in dental research as an accessible and valuable source of stem cells known as dental pulp stem cells (DPSCs). Moreover, DPSCs are going to be ideal for tissue engineering and regenerative medicine (Chalisserry et al., 2017; Honda, Sato & Toriumi, 2017; Ledesma-Martínez, Mendoza-Núñez & Santiago-Osorio, 2016). However, there are some disadvantages associated with the isolation of DPSCs, which might be directly related to the size and weight of DP tissue by limiting the number of stem cells isolated from it (Raoof et al., 2014). Besides, some studies have also focused on methods to correlate some morphometric parameter of the pulp tissue with the patient’s age. For example, Ravindra et al. (2015) observed through intra-oral radiography that the total area of the pulp decreased with age. Other authors focus on establishing regression equations using the number of cells in DP to also predict age (Hossain et al., 2017; Von Böhl et al., 2016). Conversely, until now there are no reports that relate morphometric parameters such as weight, length, width and cellular density of DP among different types of teeth and its influence on tissue quality for the isolation of DP cells. Therefore, the objective of this study was to compare the histomorphometry of DP in temporal and permanent teeth and evaluate the cell density in four regions in this tissue with to purpose of generating quantitative parameters that would have important applications in the DPSCs isolation.

Materials and Methods

Patient recruitment and tooth storage

The DP tissue was obtained from 10 deciduous central incisors, 20 permanent teeth (10 premolars, 10 third molars) and 10 supernumeraries (six mesiodents and four inferior premolar shape). An informed patient consent was obtained from patients or parents of minors. The collection of the material was performed at the Clinics of the Master in Pediatric Dentistry and Oral Surgery, Faculty of Dentistry, Autonomous University of Yucatan. The age of patients ranged from 5 to 25 years, and a slight prevalence of females (21/40, 55.5%) was noted. The protocol was approved by the Ethics Research Committee of Hideyo Noguchi Regional Research Center, Autonomous University of Yucatán (Approval Number: CIE-06-2017). The extracted deciduous teeth exhibited one-third to two-thirds root resorption with well-defined roots. After extraction, all the teeth were rinsed for 5 min in a conical tube containing phosphate-buffered saline (PBS 1X: 138 mM NaCl, 3 mM KCl, 8.1 mM Na2HPO4 and 1.5 mM KH2PO4, pH 7.4) and labeled with the donor’s age and tooth type.

Tissue removal and processing

The teeth were immersed in sterile phosphate buffer saline (PBS 1X pH 7.4), stored on ice pack and transported to the cell culture lab for sample processing. After cleaning the surface of the tooth, a vertical cut of the dental organ was performed using a rotary electric micro motor (String®) with a diamond disc (diameter: 22 mm and thickness: 0.4 mm; ATK®). During this process, constant irrigation was maintained with cold PBS 1X pH 7.4 to reduce overheating of dental tissue (Figs. 1A–1C). Thereafter, the entire pulp tissue was carefully extracted from the cavities of the tooth using a metal clamp and it was weighed using an analytical balance (Citizen CX 200) (Fig. 1D). The pulp length and width were measured with a digital Vernier caliper with a 0.01 mm calibration.

Figure 1 Cutting technique of vertical dental organ to obtain pulp.

(A) Cutting vertical of the dental organ using electric rotatory micro motor with diamond disc and irrigation with cold PBS 1X pH 7.4. (B and C) Breaking-up of the tooth along the vertical axis using a metal spatula and divided into two equal halves, one mesial and the distal. (D) Isolation of the complete DP tissue from the cavities of the tooth using a metal clamp.

Histomorphometry

The pulp tissues were fixed in 10% formaldehyde solution. Subsequently, the DP was dehydrated in increasing concentration of alcohol: 60%, 70%, 80% and 100%. After that, the DPs were embedded in paraffin and dissected in sections of 5 µm of thickness with a sliding microtome (Leica LM2500). The slices were then dewaxed and stained with hematoxylin and eosin. Four histology slides from each tooth were selected for analysis. Images of the pulp were captured through the digital microscope at a resolution of 1,280 × 720 (Leica DM750 camera MC170H) and 4–100× magnifications connected to a computer. A cell counting was performed manually under high-power (100× magnification) at four regions of each DP tissue: coronal, cervical, middle and apical. For each cell population, the number of cells was normalized to the total area of the pulp sample (325 µm × 402 µm). Afterwards, each image obtained from the histology slides were analyzed by ImageJ v1.49 Software.

Statistical analysis

A one-factor analysis of variance with Tukey’s post hoc test was used. The Pearson correlation coefficient was calculated to determine the correlation between morphometric measurements of the DP and the relationship with the patient’s age. Statistical significance was defined as p ≤ 0.05.

Results

Correlation between morphometric measurements of the dental pulp and patients’ age

A correlation study was undertaken to examine the weight, length and width of DP with patients’ age (Figs. 2A–2F). The DP weight was correlated with age in male patients, whereas significant changes (p ≤ 0.05) were not noted in females (Figs. 2A and 2B). The Pearson correlation test revealed a significant positive (p ≤ 0.05) correlation between length and width of the DP with age in both genders (Figs. 2C–2F). In general, the data showed that the DP obtained from males had a greater weight, length and width compared with that from females. The results indicate an optimal age interval in males (15–20 years) and females (20–25 years) to obtain 10 mg of DP.

Figure 2 Correlation of weight, length and width measurements of the dental pulp and its relationship with the patient’s age.

DP tissue from males and females were isolated and evaluated by (A and B) Weight (C and D) Length and (E and F) Width to determinate its correlation with the patient’s age by using the Pearson correlation coefficient. Data from all of the investigated third molar, premolars, deciduous and supernumerary are shown. The value r was calculated for all data at a significance level of p ≤ 0.05.

Comparison of morphometric measurements of dental pulp among different types of teeth

Linear data correlation and one-factor variance analyses were applied to the variables of weight, length and width (Table 1). The weight of DP was significantly increased (p ≤ 0.05) in supernumerary teeth (20.5 ± 3.56 mg) and third molars (18.7 ± 11.5 mg) from male patients compared with those from females. In contrast, the mean differences in length and width measurements of DP in the third molars were highly significant (p ≤ 0.05) compared with the other types of teeth in both genders (Table 1).

Table 1 Comparison of weight, length and width of dental pulp from different teeth between males and females.

The values of weight, width and length of the pulp of each 10 samples per tooth type were calculated. Different letters indicate significant differences between each measurement and type of teeth. p-value <0.05.

Parameters	Male	Female	Total	
Mean	SD	Mean	SD	Mean	SD	
Deciduos							
Weigth (mg)	2.40	0.011	1.7	0.0013	2.05	0.006	
Length (mm)	3.75	4.500	4.33	1.03	4.04a,b,c	2.765	
Width (mm)	1.00	0.557	1.0	0	1.0c,d	0.278	
Third molar							
Weigth (mg)	18.7	0.011	14.6	0.007	16.65	0.009	
Length (mm)	9.66	4.500	9.14	2.600	9.40b	3.550	
Width (mm)	3.66	0.557	2.57	0.786	3.115a,b,c	0.672	
Premolar							
Weigth (mg)	9.10	0.003	7.1	0.003	8.10	0.003	
Length (mm)	9.75	1.250	7.83	2.850	8.79a	2.050	
Width (mm)	1.75	0.500	1.66	0.515	1.705a	0.507	
Supernumerary							
Weigth (mg)	20.5	0.035	7.2	0.005	13.85	0.020	
Length (mm)	8.75	3.320	9.0	5.650	8.875c	4.485	
Width (mm)	1.87	0.640	1.5	0.707	1.685b,d	0.674	

Comparison of dental pulp histology and cellular density of dental pulp tissue in different types of teeth

Due the DP of the different teeth analyzed showed different morphometric measurements, we hypothesized that the density in the cell-rich zone may also be different. Thus, a histological analysis was performed in coronal, cervical, middle and apical regions of the DP to evaluate the cell density (Figs. 3A–3N). The histological evaluation revealed a three-layer structure consisting of a layer of odontoblasts (OB; Figs. 3A–3N, yellow arrowhead) with regularly arranged columnar cells in the contour of the pulp and a dispersed layer of cells in all DP samples. Conversely, in the subodontoblastic zone (SOB), a thin and cell-free layer zone (CFZ; Figs. 3A–3N, green arrowhead) was observed. The cell-rich zone (CRZ; Figs. 3A–3N, red arrowhead) showed a dense layer of cells per unit, especially in fibroblasts and undifferentiated mesenchymal cells that continue in the central zone of the pulp, in which the presence of blood vessels (BV), nerve fibers (NV) and connective tissue (CT) is highlighted (Figs. 3A–3N). The DP of the supernumeraries presented a limited presence of NV (Figs. 3I–3L) compared with the pulp obtained from the premolars, third molars and deciduous. The histology of the pulp in the coronal and cervical regions of deciduous teeth showed an irregular shape, making layers difficult to identify during staining with hematoxylin and eosin (Figs. 3M and 3N).

Figure 3 Photomicrographs of histologic sections of different regions in dental pulp from premolars, third molar, supernumerary and deciduous teeth.

Microscopic image demonstrating a typical tissue from (A–M) coronal, (B–N) cervical, (C–K) middle and (D–L) apical regions of DP in different teeth. All sections were stained with hematoxylin and eosin. OB, odontoblast layer (yellow arrowhead); D, dentin; CFZ, cell-free zone (green arrowhead); CRZ, cell-rich zone (red arrowhead); BV, blood vessels; NF, nerve fibers; D, dentin; P, pulp and CT, connective tissue. UD: denotes Not determined, this regions are not found in deciduous teeth (O and P).

A greater cellular density was observed in the coronal region of deciduous and premolar teeth compared with supernumerary and third molars (Table 2). In contrast, a high cell density was observed in the apical region of supernumeraries and premolars compared with third molar teeth (Table 2). Together, these results indicate that the apical region of the DP in supernumerary and premolar teeth potentially represents the ideal location to obtain cells.

Table 2 Cell density values in specific regions of dental pulp tissue from different teeth.

All values were calculated from three histological sites for each one of the regions of DP among the different teeth expressed as the mean ± standard deviation. (a, b and c) indicate significant differences in each region of DP and the type of tooth. (A, B, C, D and E) indicate significance between different tooth and each region of DP, p ≤ 0.05. UD: denotes Not determined, this regions are not found in deciduous teeth.

Teeth	Cell density (cel/mm2)	
Coronal	Cervical	Middle	Apical	Mean total	
Premolar	40.90 ± 7.46	36.80 ± 9.09	25.80 ± 8.07	51.60 ± 10.12	38.78 ± 12.57a	
Third molar	19.20 ± 4.42	19.40 ± 4.43	12.20 ± 5.22	30.30 ± 2.00	20.28 ± 7.70a,b,c	
Supernumerary	39.70 ± 12.63	38.50 ± 10.16	24.90 ± 5.71	61.00 ± 10.08	41.03 ± 16.21b	
Deciduous	47.30 ± 14.71	46.80 ± 14.74	UD	UD	47.05 ± 14.34c	
Mean total	36.78 ± 14.77A,B	35.38 ± 14.12C,D	20.97 ± 8.87A,C,E	47.63 ± 15.34B,D,E		

Discussion

Knowledge of DP histomorphometry in teeth is important for identifying the tooth type that could provide the best source of cells. Several researchers have focused on obtaining DPSCs from exfoliated deciduous or permanent third molars because stem cells from these teeth exhibit a high proliferation capacity (Daud et al., 2016; Shekar & Ranganathan, 2012). It is therefore necessary to identify other source of DPs with quality cellular characteristics for the isolation of cells. In the literature, reports on morphometric and histological measurements of the DP regions among temporary and permanent teeth are quite limited. Besides, pulp weight is a parameter that few authors have considered to obtain DPSCs (Alsulaimani et al., 2016; Kellner et al., 2014; Singh et al., 2016). Our research group considers that DP weight could be important as a starting point for the isolation of DPSCs. The results of the present study show that the DP weight was greater in the supernumerary and third molars teeth in male patients compared with female patients (Table 1). The weight observed in supernumerary (mesiodents) teeth was may be due to the dense fibrous DP, which could indicate a greater amount of organic substance in this tooth type. The third molar DP of females exhibited an increased weight compared with males likely because female DP was obtained from upper teeth with fused roots. This characteristic facilitated the procurement of a larger homogenous pulp tissue compared with males whose third molars were primarily from the mandible and with separated roots. In Méndez & Zarzoza (1999) reported a mean weight pulp of 13.10 ± 4.33 mg in eight premolars fragmented with a hammer. In our study, we used a vertical cutting method to obtain DP from all types of teeth. This variation in weight could be explained by the fact that the fractionation by impact could lead to the loss of pulp tissue. In addition, some premolars had separated roots, so the size of the pulp tissue may reduce.

On the other hand, it is also important to note that the DP also undergoes age-related changes, and several studies have focused on aging. Recently, Kellner et al. (2014) determined that the relation between pulp vs. hard tooth tissue in third molar decreases with aging. This finding was also observed in our study (Fig. 2). Unfortunately, no reports on deciduous pulp weight were found to compare our observations.

The length of DP was three-fold higher in permanent compared with deciduous teeth (Table 1) because deciduous teeth showed a physiological reabsorption that does not occur in permanent teeth. Regarding DP width, measurements of deciduous and permanent teeth are obtained using radiological techniques, such as pericapical X-rays or orthopantomography. For example, Kazmi, Anderson & Liversidge (2017) used conventional radiology to measure the mesio-distal crown width of deciduous teeth in males and females, revealing no significant differences, which was consistent with our findings (Table 1). However, there are no reports about the in vitro length of the pulp for comparison with our results. On the other hand, the cell-rich zone contains progenitor cells that exhibit plasticity and pluripotency. For instance, Lizier et al. (2012) indicated that DPSCs are located in multiple niches, which are associated with capillaries and the nerve network of the central region in the CRZ and in the outer layer of pulp tissue (Graziano et al., 2008; Pagella et al., 2015). We observed that the apical region of the DP of the supernumerary teeth, showed a higher cell density compared to the other permanent teeth. Interestingly, in the total analysis of cell density, the supernumerary teeth have similar cellular density compared to deciduous teeth (Table 2). Although the pulp tissue of the deciduous teeth was smaller, this pulp exhibited the highest cell density compared with the other dental organs (Table 2). This finding could be explained because the coronal and apical regions are held together after a physiological resorption in deciduous teeth.

Gronthos et al. (2002) showed that the DP derived from lower deciduous central incisors contains a large number of cells able to form adherent colonies similar to mesenchymal stem cells in in vitro culture. In contrast, the small number of cells available for isolation due to the size of the pulp, especially in exfoliated deciduous teeth represents a potential problem with obtaining DPSCs (Marrelli et al., 2018). Thus, one of the advantages of supernumerary teeth for the isolation of cells is that these teeth are extracted at an early age, which apparently retain embryogenic characteristics as demonstrated for another source of stem cells from the oral cavity (Dunaway et al., 2017). These results indicate that supernumerary teeth in patients between 5–20 years of age have the best morphometric parameters. However, the determination of the biological properties such as proliferation and differentiation potential of the isolated cells in the different regions of this tissue requires further studies.

Conclusions

In this study, in vitro histomorphometric comparison and cellular density of the DP from temporary and permanent teeth of patients from 5 to 25 years of age were addressed. It was shown that supernumerary DP has the best morphometric parameters and its cell density is comparable to that of deciduous tooth pulp. This phenomenon has not been described before and could have important applications in the isolation of stem cells in this tissue.

Supplemental Information

Supplemental Information 1 Raw data parameter analysis.

Click here for additional data file.

Supplemental Information 2 Raw Data of pulp parameters.

Click here for additional data file.

Supplemental Information 3 Raw data cell density in different teeth.

Click here for additional data file.

We are grateful for the technical support of M.C. Angela Ku González in the treatment of dental pulp tissue for histological analysis. The authors also acknowledge the support of the Seeding Labs Program for manuscript editing.

Additional Information and Declarations

Competing Interests

Author Contributions

Human Ethics

Data Availability

The authors declare that they have no competing interests.

Marytere Guerrero-Jiménez conceived and designed the experiments, performed the experiments, prepared figures and/or tables, authored or reviewed drafts of the paper, approved the final draft.

Geovanny I. Nic-Can conceived and designed the experiments, performed the experiments, prepared figures and/or tables, authored or reviewed drafts of the paper, approved the final draft.

Nelly Castro-Linares conceived and designed the experiments, analyzed the data, authored or reviewed drafts of the paper, approved the final draft.

Fernando Javier Aguilar-Ayala analyzed the data, contributed reagents/materials/analysis tools, authored or reviewed drafts of the paper, approved the final draft.

Michel Canul-Chan analyzed the data, prepared figures and/or tables, authored or reviewed drafts of the paper, approved the final draft.

Rafael A. Rojas-Herrera analyzed the data, contributed reagents/materials/analysis tools, authored or reviewed drafts of the paper, approved the final draft.

Ricardo Peñaloza-Cuevas conceived and designed the experiments, performed the experiments, analyzed the data, contributed reagents/materials/analysis tools, authored or reviewed drafts of the paper, approved the final draft.

Beatriz A. Rodas-Junco conceived and designed the experiments, performed the experiments, analyzed the data, contributed reagents/materials/analysis tools, prepared figures and/or tables, authored or reviewed drafts of the paper, approved the final draft.

The following information was supplied relating to ethical approvals (i.e., approving body and any reference numbers):

The Ethics Research Committee of Hideyo Noguchi Regional Research Center, Autonomous University of Yucatán granted Ethical approval to carry out the study (Approval Number: CIE-06-2017).

The following information was supplied regarding data availability:

The raw data is available in the Supplemental Files.

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
