# Peer review of "In vitro histomorphometric comparison of dental pulp tissue in different teeth"

_PeerJ, doi:10.7717/peerj.8212_

## Round 0.1 · original submission · Major Revisions

Please address the reviewers concerns on a point by point basis and resubmit the manuscript. In particular, the critical comments of Reviewer 1 must be adequately addressed.

Reviewer 1 ·

Basic reporting

No comment

Experimental design

No comment

Validity of the findings

No comment

Additional comments

This study investigated the histomorphometric properties of dental pulp tissues obtained from different tooth types. The effect of age and sex was also reported. The results showed differences in the weight and also cell density of the dental pulps among the groups. The authors suggest using these parameters to predict the availability and quality of dental pulp stem cell isolation. There are some concerns regarding the aim and methodology of the present study.

There has been a lack of standardization in tissue samples in the present study. It is hard to standardize the tissue volume when determining the cell number due to the differences in pulpal volume in different tooth types and different donors. In addition, the authors did not include the scale bar to the histological slides and the image resolutions are too low for the evaluation. Regarding the isolation of dental pulp stem cells, undoubtedly younger patients are a more suitable candidate due to the decreased pulp volume and cellular contents by aging. Although there is a decreased cellular intensity by aging shown by the previous and present results, the percentage of DPSCs might not be changed. In this point, the isolation of DPSCs and also the evaluation of their biological properties such as proliferation and differentiation properties are important. Previously, Bressan et al (Plos One, 2012) reported that DPSCs isolated from 16-66-year-old patients could retain the proliferation and differentiation potential. In the present study, the authors examined the dental pulp tissues derived from young donors up to 25 years, which any difference regarding the percentage and properties of DPSCs would not be expected.

Regarding the correlation between the histomorphometric properties of the dental pulp tissues and the isolation of cells would be further investigated with the determination of biological properties of the isolated DPSCs. In this point, the statement cannot be made by the results of the current study.

·

Basic reporting

The manuscript entitled "In vitro histomorphometric comparison of dental pulp tissue in different teeth" reports a novel methods that demonstrated a significant positive correlation noted between length and width of the dental pulp with age in both males and females. I believe the manuscript still requires some minor revision before further consideration for publication.

Experimental design

The authors has described the experimental design very well. However some minor changes that has been mentioned in the general comments are needed to be completed.

Validity of the findings

The current manuscript demonstrates that the mean dental pulp weight of supernumerary and third molar teeth was greater than deciduous and premolar teeth and finally, the histological analysis revealed that the apical portions of dental pulp in supernumerary and premolar teeth have the highest cell density and provide a greater number of cells for in vitro culture. Overall, the current finding has important applications in the dental pulp stem cells isolation.

Additional comments

1. The author should change the abbreviation “DP” in the results section of the abstract to the full word and use the word first time in full and then only in abbreviation.

2. At line 55, the author has used the cells name in full words as well as in abbreviation “Dental pulp stem cells (DPSCs)” and the same format has repeated at line 141. The same mistakes have repeated throughout the manuscript. Please follow the proper format.

3. Materials and Methods: Tissue removal and processing section, “Tooth samples were longitudinally cut using a micromotor”. The authors need to confirm whether the pulp has been little bit damaged or remain intact while using the microcutter? Please mention the diameter of the cutter.
4. Figure 2A, the DP weight of only one male is more than 100mg whereas all the other DPs weights are showing less than 30mg. Why there is a huge difference in weight in DP weight of male? Explain them precisely.

5. Figure 2 legends: “Statistical significance was defined as p ≤ 0.05”. However there is no symbol representing statistical significance in the figure. Add the symbols.

6. Figure 3 legends: “Values were calculated from three slices of each of the 10 teeth in each group and are expressed as the mean ± standard deviation. *P ≤0.05”. Add the symbol * in the figures?

7. In “Figure caption section” more detail description are needed to add for making clear easily follow.
8. There are grammar errors in the manuscript and the tenses are confused. Therefore, the manuscript should be thoroughly edited.
9. In “Table caption section” more detail description are needed to add for making clear easily follow.
10. I advise the authors to correct the grammatical errors because there are a large number of grammatical mistakes or odd turns of phrase, and too many for a reviewer to list.

---

## Round 0.2 · Minor Revisions

Please make the changes suggested by the reviewer and provide a response to the reviewer on a point by point basis.

Reviewer 1 ·

Basic reporting

Thank the authors for their revision according to the previous comments, but still, some revisions are required.

Experimental design

No comment

Validity of the findings

No comment

Additional comments

Page 7, line 61, please remove the sentence “This fact motivates our research and concomitantly ensures the novelty of the results”.

Page 8, line 77, please add PBS brand.

In the results section, please add the P values (p<0.05 or p>0.05) after each sentence.

For the histological images, please label dentin and pulpal side of the samples. The images are still not well enough for the resolution.

In Table 2, the statistical comparisons should be also performed among coronal, cervical, middle and apical part of each tooth type. According to that, the ideal cell location might not be limited to the apical region of supernumerary and premolar teeth.

·

Basic reporting

The manuscript entitled "In vitro histomorphometric comparison of dental pulp tissue in different teeth" reports a novel methods that demonstrated a significant positive correlation noted between length and width of the dental pulp with age in both males and females. I believe that the manuscript in the current format have adequate significance to be of interest to the readers of the journal.

Experimental design

The paper is well presented and the experimental design used for the study is acceptable.

Validity of the findings

The current manuscript demonstrates that the mean dental pulp weight of supernumerary and third molar teeth was greater than deciduous and premolar teeth and finally, the histological analysis revealed that the apical portions of dental pulp in supernumerary and premolar teeth have the highest cell density and provide a greater number of cells for in vitro culture. Overall, the current finding has important applications in the dental pulp stem cells isolation.

---

## Round 0.3 · accepted · Accept

You have addressed the issues raised by the reviewers in a reasonable manner. This has significantly improved the quality of the manuscript.